# Projection of National Carbon Footprint in Japan with Integration of LCA and IAMs

**Yuki Ichisugi [1,\*], Toshihiko Masui [2], Selim Karkour [1] and Norihiro Itsubo [1]**

[1]   Graduate School of Environmental and Information Studies, Tokyo City University,
     Yokohama 224-8551, Japan; g1893103@tcu.ac.jp (S.K.); itsubo-n@tcu.ac.jp (N.I.)

[2]   National Institute for Environmental Studies, Tsukuba 305-0553, Japan; masui@nies.go.jp

\*   Correspondence: g1793102@tcu.ac.jp; Tel.: +81-45-910-2930

**Abstract:** In order to achieve target greenhouse gas (GHG) emissions, such as those proposed by each country by nationally determined contributions (NDCs), GHG emission projections are receiving attention around the world. Generally, integrated assessment models (IAMs) are used to estimate future GHG emissions considering both economic structure and final energy consumption. However, these models usually do not consider the entire supply chain, because of differences in the aims of application. In contrast, life cycle assessment (LCA) considers the entire supply chain but does not cover future environmental impacts. Therefore, this study aims to evaluate the national carbon footprint projection in Japan based on life cycle thinking and IAMs, using the advantages of each. A future input–output table was developed using the Asia-Pacific integrated model (AIM)/computable general equilibrium (CGE) model (Japan) developed by the National Institute for Environmental Studies (NIES). In this study, we collected the fundamental data using LCA databases and estimated future GHG emissions based on production-based and consumption-based approaches considering supply chains among industrial sectors. We targeted fiscal year (FY) 2030 because the Japanese government set a goal for GHG emissions in 2030 in its NDC report. Accordingly, we set three scenarios: FY2005 (business as usual (BAU)), FY2030 (BAU), and FY2030 (NDC). As a result, the carbon footprint (CFP) in FY2030 will be approximately 1097 megatons of carbon dioxide equivalent (MtCO$_2$eq), which is 28.5% lower than in FY2005. The main driver of this reduction is a shift in energy use, such as the introduction of renewable energy. According to the results, the CFP from the consumption side, fuel combustion in the use stage, transport and postal services, and electricity influence the total CFP, while results of the production side showed the CFP of the energy and material sectors, such as iron and steel and transport, will have an impact on the total CFP. Moreover, carbon productivity will gradually increase and FY2030 (NDC) carbon productivity will be higher than the other two cases.

**Keywords:** dynamic evaluation; input–output analysis; integrated assessment model; computable general equilibrium; carbon footprint; inventory database

---

## 1. Introduction

Following the Intergovernmental Panel on Climate Change (IPCC) special report on global warming of 1.5 °C published in 2018 [1], evaluation of greenhouse gas (GHG) emissions has received more attention around the world. In addition, estimation of environmental burden projections based on socioeconomic parameters such as shared socioeconomic pathways (SSPs) is also becoming important.

Life cycle assessment (LCA) is a method to quantitatively assess the environmental impacts of products and services "from cradle to grave," that is, the entire supply chain. The scope is extended to a variety of directions for assessment of organizations and analysis of consumer consumption patterns and lifestyles at a national or international level [2].

In particular, LCA at the national level has received attention as applied to environmental policy-making. The EXIOBASE consortium published a report about the environmental footprints of nations [3]. Carbon, water, land, and material footprints were calculated considering final consumption, using the multi-regional input–output (MRIO) table EXIOBASE 2.1. That calculation method was based on input–output analysis (IOA), which can be used to analyze both production-based and consumption-based approaches.

In the production-based approach, environmental burden is classified from the production side, such as manufacturers and electric power companies. On the other hand, in the consumption-based approach, environmental burden is allocated according to the final products to be consumed, such as by households. Generally, in evaluating the environmental footprint, the consumption-based approach is chosen because the total demand within a nation cannot be satisfied only with domestic industries. In addition, these results contribute to promoting the purchase of low environmental impact products. Therefore, the environmental footprint is calculated following the consumption-based approach considering the entire supply chain involving foreign trade. Some governments have been following this approach, such as the British government, which has published the "UK's Carbon Footprint" report [4] in recent years. In this report, the carbon footprint (CFP) is divided into three categories: GHGs embedded in imported goods and services, GHGs from UK-produced goods and services consumed by UK residents, and GHGs generated directly by UK households.

Following the scope of the study, there are several methods to determine the carbon footprint, which are summarized in Figure 1. As shown in this figure, the input–output based approach is usually chosen when considering larger scales such as national or global. The multi-regional input–output (MRIO) table, for example, was chosen to estimate the burden caused by imports and exports and to conduct a comprehensive evaluation of the supply chain with consideration of the final demand [5]. An LCA database, Embodied Energy and Emission Intensity Data for Japan Using Input–Output Tables (3EID), based on the national input–output table, was developed in Japan by Nansai [6]. In this method, direct GHG emissions per monetary unit (Japanese yen) are calculated based on the fuel consumption and resource input for each industrial sector, and the indirect burden through the supply chain is also quantified by using the Leontief inverse matrix [7]. This method was applied in Japan to determine, for instance, the carbon footprints of principal cities [8] and medical services [9].

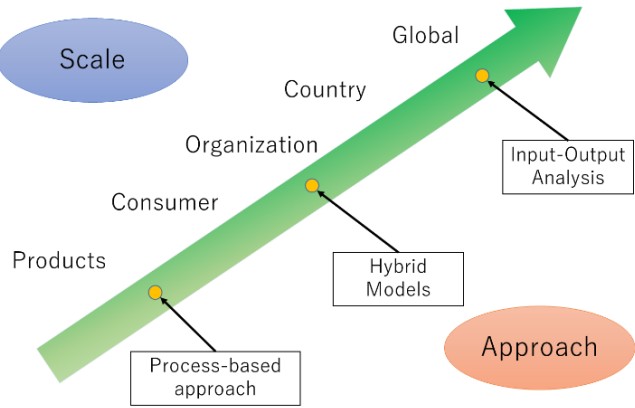

**Figure 1.** Relationship between scale of carbon footprint and calculation approach (adapted from [10]).

However, these approaches are based on data published for a single year (for example, 2011 in Japan) and do not take into account annual socioeconomic evolution, and therefore they cannot be used to estimate future projections.

Integrated assessment models (IAMs) can overcome this limitation, based on approaches that integrate knowledge from several domains such as natural science and economics into a single framework. IAMs were used, for example, to establish GHG emission projections based on targets set up by each government [11]. Generally, the input of such models includes the introduction of

carbon-capture storage (CCS) technology, renewable energy systems, and improvement of technology performance in each industrial sector.

There are several types of global IAMs, as reviewed in Table A1 of the Appendix A. One of them is the Asia-Pacific integrated model (AIM) [12] developed by the National Institute for Environmental Studies (NIES). This model can be used for estimating future GHG emissions and assessing policy options, including the carbon budget in the Asia-Pacific area, as explained in [13], for instance, where the energy mix in 2030 is suggested considering Japan's intended nationally determined contribution (INDC) [14]. IAMs can therefore estimate both future economic structure and future GHG emissions from the final energy demand at either a global or national level. They can be used to take measures against climate change, as described in the different IPCC reports.

However, while final total GHG emissions could be estimated in previous studies in Japan [15], indirect GHG emissions from the entire supply chain were not specifically considered. In addition, most of the global and country-level evaluations conducted only included a few sector classifications [16]; in order to address specific recommendations to policy-makers, a much more detailed industrial approach needs to be considered to visualize a relationship among industrial sectors.

Moreover, there is another point that has not been considered: Even if some studies focused on indirect emissions from the supply chain using 3EID in the past (e.g., 2005), the future relationship between sectors has not yet been considered. By using IAMs, it is possible to estimate future indirect emissions from the supply chain including trades.

Therefore, this study aims to develop a dynamic evaluation and projection of carbon footprint as an environmental burden in the future based on life cycle thinking using the advantages of both LCA and IAMs. More specifically, AIM Japan [17] was used, as it contains specific information concerning the industrial sectors in the country in comparison with other AIMs. Another reason is that the social accounting matrix (SAM) introduced in the model shows good affinity with the input–output table. This study uses AIM Japan to estimate the future input–output table and calculate the CFP in 2030 to provide advice to policy-makers but also give information to citizens.

## 2. Materials and Methods

### 2.1. Asia-Pacific Integrated Model (AIM)

The calculation method of the carbon footprint (CFP) is described in Figure 2. We estimated the social accounting matrix (SAM) including an input–output (IO) table for the future using the results from the computable general equilibrium (CGE) model, AIM/CGE (Japan) [17]. The model has 43 commodities and 49 sectors, as shown in Table 1. It also includes the aggregated GHG mitigation technologies, which are summarized in the results from AIM/Enduse (Japan) [18]. The estimated SAM reflects the additional costs of GHG mitigation technologies.

In this study, fiscal year (FY) 2005 is treated as a base year, and FY2030 is set to be a target year in order to assess the nationally determined contribution (NDC) in Japan [19]. As benchmark data, the AIM/CGE (Japan) uses the 2005 Japanese IO table (JIOT) [20] released by the Ministry of Internal Affairs and Communications and the GHG emissions data from the National Greenhouse Gas Inventory Report (NIR) of Japan [21]. Economic activity from FY2005 to FY2030 is calculated based on several assumptions such as GDP, population, electricity supply mix, and target GHG emission reduction. Table 2 shows the statistical data and assumptions of GDP [22], population [23], and GHG emissions [24] in FY2005 and FY2030. Table 3 shows the electricity supply mix in FY2005 and FY2030 based on the NDC. According to the NDC, the share of renewable energy will be increased to about 23% in FY2030 [19,24]. By comparison, that increase was assumed to be around 30% in the Japanese government projections. This difference is caused by the specificity of the AIM/CGE (Japan) model, in which each industrial sector chooses its electricity source considering the cost performance. According to the Japanese NDC report [22], GHG emissions in FY2030 will be reduced by approximately 25.4% compared to FY2005. In addition, the government will also introduce carbon budget measures.

The BAU scenario in FY2030 considers GDP, population, and energy mix. The NDC scenario in FY2030 also considers GDP, population, and energy mix as exogenous parameters, but also available GHG mitigation technologies and emission reduction targets. All scenarios use the same emission factors issued from the NIR published in Japan [21].

In AIM/CGE (Japan), one of the results is the SAM for each year from FY2005 to FY2030. Based on the estimated SAM, the input–output table for the future was calculated. Then, we calculated the CFP projection using the IOA method. In the CFP analysis, 40 sectors are formulated as basic sectors. To be consistent with CGE results, the CFP analysis uses the 2005 Japanese IO table (JIOT) [20], and the direct GHG emissions data from NIR of Japan in FY2005 [21]. The validity of the GHG database was confirmed by comparing embodied energy and emission intensity data for Japan using the input–output table (3EID) developed by NIES [25], as shown in Section 4.1. 3EID contains direct and indirect GHG emissions of each sector incorporated in the 2005 JIOT.

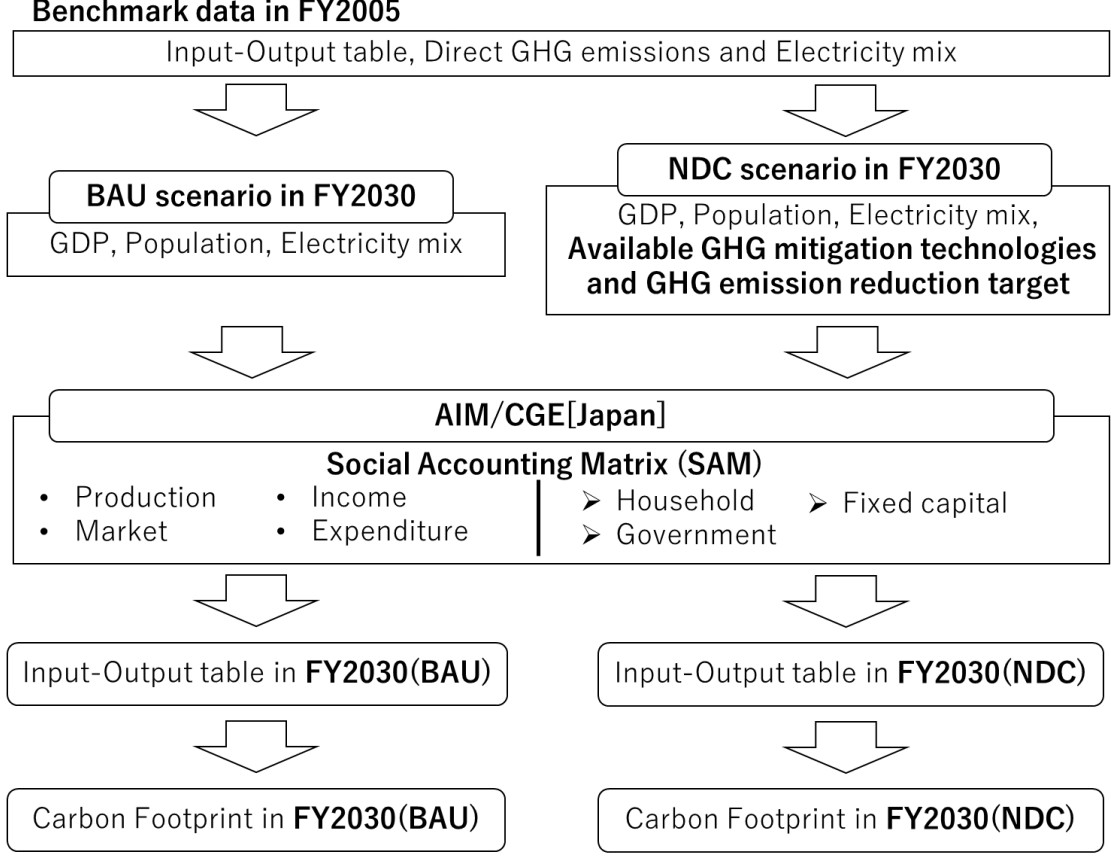

**Figure 2.** Flowchart to estimate carbon footprint (CFP) in 2030. GHG, greenhouse gas; BAU, business as usual; GDP, gross domestic product; NDC, nationally determined contribution; AIM/CGE, Asia-Pacific integrated model/computable general equilibrium.

**Table 1.** Sector classification.

| | Goods/Services (Row) | | Goods/Services (Column) | | Goods/Services (Row) | | Goods/Services (Column) |
|---|---|---|---|---|---|---|---|
| 1 | Agriculture, forestry, and fishery | 1 | Agriculture, forestry, and fishery | | | 24p | Private power generation |
| 2 | Mining | 2 | Mining | | | 24n | Nuclear power generation |
| 3c | Coal mining | | | | | 24tc | Coal thermal power generation |
| 3o | Petroleum | 3 | Coal mining, petroleum, and natural gas | | | 24to | Petroleum thermal power generation |
| 3g | Natural gas | | | 24 | Electricity | 24tg | Gas thermal power generation |
| 4 | Beverages and foods | 4 | Beverages and foods | | | 24h | Hydraulic power generation |
| 5 | Textile products | 5 | Textile products | | | 24s | Solar power generation |
| 6 | Pulp and paper | 6 | Pulp and paper | | | 24w | Wind power generation |
| 7 | Chemical products | 7 | Chemical products | | | 24g | Geothermal power generation |
| 8m | Petroleum products (motor vehicle) | 8 | Petroleum products | | | 24b | Biomass power generation |
| 8o | Petroleum products (other) | | | 25 | Gas supply | 25 | Gas supply |
| 9 | Coal products | 9 | Coal products | 26 | Steam and hot water supply | 26 | Steam and hot water supply |
| 10 | Plastic and rubber | 10 | Plastic and rubber | 27 | Water supply | 27 | Water supply |
| 11 | Ceramic, stone, and clay | 11 | Ceramic, stone, and clay | 28 | Waste management service | 28 | Waste management service |
| 12 | Iron and steel | 12 | Iron and steel | 29 | Commerce | 29 | Commerce |
| 13 | Nonferrous | 13 | Nonferrous | 30 | Finance and insurance | 30 | Finance and insurance |
| 14 | Metal products | 14 | Metal products | 31 | Real estate | 31 | Real estate |
| 15 | General-purpose machinery | 15 | General-purpose machinery | 32 | Transport and postal services | 32 | Transport and postal services |
| 16 | Production machinery | 16 | Production machinery | 33 | Information and communications | 33 | Information and communications |
| 17 | Business-oriented machinery | 17 | Business-oriented machinery | 34 | Public administration | 34 | Public administration |
| 18 | Electronic components | 18 | Electronic components | 35 | Education and research | 35 | Education and research |
| 19 | Electronic machinery | 19 | Electronic machinery | 36 | Medical, health care, and welfare | 36 | Medical, health care, and welfare |
| 20 | Information and communication electronics products | 20 | Information and communication electronics products | 37 | Miscellaneous nonprofit service | 37 | Miscellaneous nonprofit service |
| 21 | Transportation equipment | 21 | Transportation equipment | 38 | Business services | 38 | Business services |
| 22 | Miscellaneous manufacturing products | 22 | Miscellaneous manufacturing products | 39 | Personal services | 39 | Personal services |
| 23 | Construction | 23 | Construction | 40 | Office supplies and activities not elsewhere | 40 | Office supplies and activities not elsewhere |

**Table 2.** Assumption of socioeconomic conditions and GHG emissions in NDC.

| Index | FY2005 | FY2030 | Ratio (%) |
|---|---|---|---|
| Real GDP (trillion JPY at 2005 price) | 507 | 711 | 40.2 |
| Population (million persons) | 128 | 117 | −8.6 |
| GHG emissions (million tons $CO_2$eq) | 1397 | 1042 | −25.4 |

**Table 3.** Target of energy mix in this study [21].

| Index | FY2005 (BAU) | FY2030 (BAU) | FY2030 (NDC) |
|---|---|---|---|
| Final energy consumption (M kl) | 410 | 326 | 326 |
| Total power generation (billion kWh) | 1149 | 1056 | 1056 |
| Coal (%) | 24.1 | 30.4 | 24.7 |
| LNG (%) | 22.0 | 15.5 | 18.9 |
| Oil (%) | 11.8 | 0.00 | 1.20 |
| Nuclear (%) | 27.2 | 18.4 | 18.7 |
| Renewable energy (%) | 8.4 | 30.1 | 30.6 |
| Hydro (%) | 6.5 | 8.7 | 8.9 |
| Solar (%) | | 12.5 | 12.7 |
| Wind (%) | | 2.2 | 2.3 |
| Geothermal (%) | 1.0 | 1.0 | 1.0 |
| Biomass (%) | | 5.7 | 5.8 |
| Private (%) | 6.6 | 5.7 | 5.8 |

### 2.2. Inventory Database Based on Input–Output Analysis

This study developed a GHG intensity database using IOA. The economic input–output (EIO) model was developed by Leontief [26] and is generally used as a quantitative model for life cycle assessment (LCA). IOA can analyze the flow of products and services between economic sectors in addition to final demand.

### 2.2.1. Matrix of Direct Input Coefficients

The direct input coefficient is the ratio of intermediate demand inputs (sales from sector *i* to sector *j*), $X_{ij}$. The set of input coefficient of all economic sectors is expressed in the square matrix A ($n \times n$), which is called the direct input coefficient matrix:

$$A = a_{ij} = X_{ij}/X_j \quad (i, j = 1, \ldots, n) \tag{1}$$

Similar sectors are aggregated or merged for all individual outputs into one aggregated output.

### 2.2.2. Direct GHG Emissions

The GHG coefficient matrix (*D*) is the extension of the direct input coefficient matrix, defined as follows:

$$D = d_{kj} = D_{kj}/X_j \quad (k = 1, \ldots, m; \; j = 1, \ldots, n) \tag{2}$$

where *D* is a $k \times j$ matrix and $d_{kj}$ is elementary flow *k* per monetary output of sector *j*.

In this study, six elementary flows were covered as GHG emissions: carbon dioxide, methane, nitrous oxide, hydrofluorocarbons (HFCs), perfluorocarbons (PFCs), and sulfur hexafluoride.

### 2.2.3. Calculation of Environmental Intensity

GHG intensity $e_k$ is defined as follows:

$$e_k = D(I - A)^{-1} \quad (k = 1, \ldots, 6) \tag{3}$$

where *A* is the direct input coefficient matrix (calculated by dividing the industry-by-industry direct requirements of sectoral inputs by the sectoral output), *I* is an identity matrix, and *D* is the direct GHG coefficient matrix. $(I - A)^{-1}$ is the Leontief inverse matrix, which considers the ripple effect of the economy. GHG intensity $e_k$ therefore includes both direct and indirect GHG emissions.

2.2.4. Calculation of CFP

CFP $E_k$ is defined as follows:

$$E_k = \sum_{j=1}^{n} e_{kj} f_j + E'_k \quad (k = 1, \ldots, 6) \tag{4}$$

where $f_j$ is final demand (household, fixed capital, government, and stocks) and $E'_k$ is fuel combustion. Generally, the CFP is calculated by including imports based on consumption. In this study, intermediate sectors of IO include imports based on the assumption that these imports are produced using the same technologies as those used in Japan. Thereby, imports sectors are written as negative values in IO. Thus, $f_j$ in formula (4) does not include import value. Formula (4) covers cradle-to-grave in the life stage of products and services, therefore it excludes the use stage. In this case, we should take into account the impact of the use stage as activity origin. $E_k$ is calculated for multiple GHG intensity and final demand. In addition, the fuel combustion is also considered.

Moreover, we analyzed the production and consumption sides to grasp which industrial sector has impact on CFP among other sectors and which sector demand influences other sectors. Each CFP is defined as follows:

$$CFP_{production\ side} = \sum_{j=1}^{n} d_{kj} X_j + E'_k \quad (k = 1, \ldots, 6) \tag{5}$$

$$X_j = L \cdot f_j \tag{6}$$

$$CFP_{consumption\ side} = \sum_{j=1}^{n} Y_{kj} f_j + E'_k \quad (k = 1, \ldots, 6) \tag{7}$$

$$Y_{kj} = d_{kj} \cdot L \tag{8}$$

where $d_{kj}$ is direct GHG emissions in each sector, $L$ is the Leontief inverse matrix, and $f_j$ is final demand. $X_j$ is calculated by $L$ and $f_j$; $Y_{kj}$ is calculated by $d_{kj}$ and $L$. The difference between formulas (5) and (7) is just the order of calculation, that is, they use the same parameters. Formula (5) expresses the CFP from the production side, where the environmental burden is caused by production sectors. Formula (7) expresses the CFP from the consumption side, where the environmental burden is caused by purchasing sectors.

2.2.5. Carbon Productivity

Carbon productivity is defined as follows:

$$Carbon\ productivity_n = \frac{f_n}{E_n} \quad (n = 1, \ldots, 40) \tag{9}$$

where $E_n$ is the CFP calculated by formula (4) by sector, $f_n$ is the final demand, and $n$ is the number of intermediate sectors. Intensity is estimated by dividing $f_n$ by $E_n$. This value is regarded as carbon productivity because the relationship between economic indicators and environmental burden in each industry is visualized.

Carbon productivity is a very effective index when considering both environmental protection and economic development: sectors that have a strong influence on economic growth and low emissions as well as sectors that have a small influence on economic growth and high emissions can both be highlighted.

## 3. Results

### 3.1. Carbon Footprint Projections

The total CFP projections based on the three scenarios are shown in Figure 3, and the population and CFP per capita are shown in Table 4. The CFP in FY2005 is estimated to be approximately

1535 million megatons of carbon dioxide equivalent (MtCO$_2$eq) and in FY2030 (BAU) it is estimated to be approximately 1251 MtCO$_2$eq. Therefore, according to the results, the CFP in FY2030 (BAU) is 18.5% lower than in FY2005. Focusing on the final demand sectors: the CFPs of household, fixed capital, and government are respectively 12%, 35%, and 21% lower than in FY2005. Between FY2005 and FY2030 (BAU), GDP and population are the main differences. Especially in this study, GDP in FY2030 (BAU) is estimated to be 40.2% higher than in FY2005 and total population 7% lower than in FY2005.

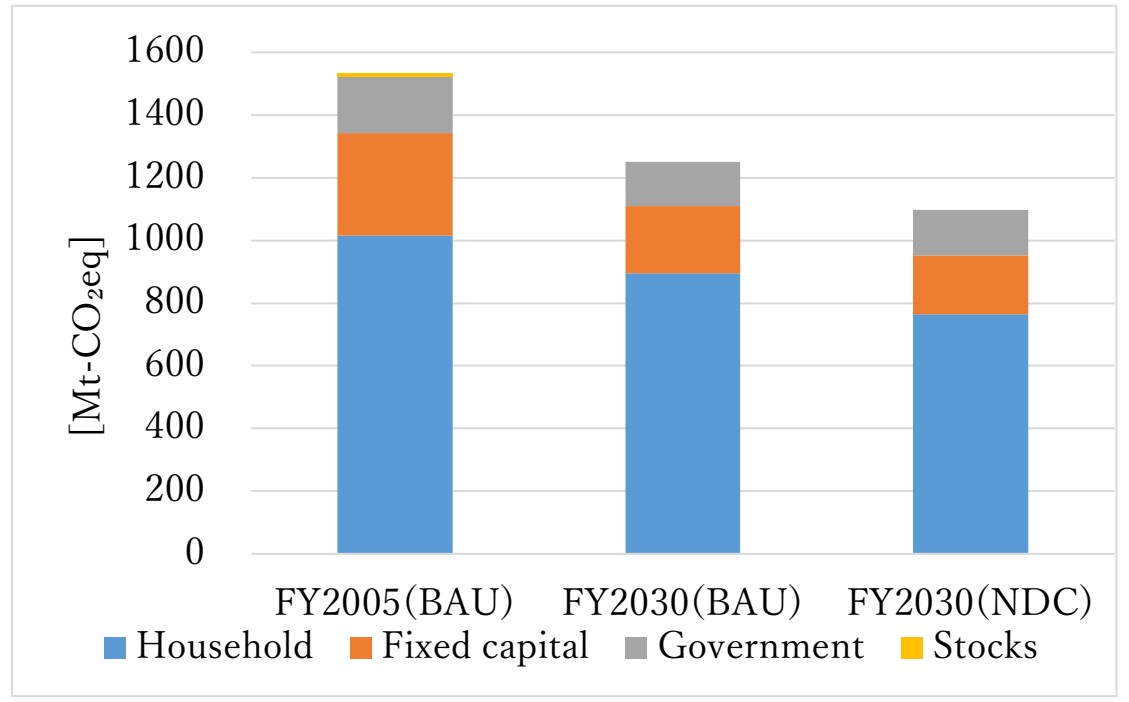

**Figure 3.** Total CFP projections.

**Table 4.** Total CFP and population in 2030.

| Index | FY2005 (BAU) | FY2030 (BAU) | FY2030 (NDC) |
|---|---|---|---|
| CFP (megatons of carbon dioxide equivalent (MtCO$_2$eq)) | 1535 | 1251 | 1098 |
| Population (million persons) | 128 | 117 | 117 |
| CFP per capita (tCO$_2$eq) | 12.0 | 10.6 | 9.34 |

In contrast, the CFP in FY2030 (NDC) is estimated to be approximately 1097 MtCO$_2$eq, which is 28.5% lower than FY2005 and 12.2% lower than FY2030 (BAU). This is attributed not only to the shift of GDP and population composition, but also to global warming countermeasures. In particular, the difference between FY2030 (BAU) and FY2030 (NDC) is caused by differences in actions to mitigate global warming effects. The CFP per capita is estimated as follows: 12.0 tCO$_2$eq in FY2005, 10.6 tCO$_2$eq in FY2030 (BAU), and 9.34 tCO$_2$eq in FY2030 (NDC). The CFP per capita in FY2030 is lower due to improved environmental performance and decreased population. By comparison to total GHG emissions, which will be reduced by 25.4% in the NDC scenario, the CFP in FY2030 (BAU) will not achieve that goal. However, from our results, the CFP in FY2030 (NDC) will be able to achieve that goal.

Figure 4 shows the industrial-level breakdown of CFP from the viewpoint of the production and consumption sides. On the production side, electricity; fuel combustion; coal mining, petroleum, and natural gas; transport and postal services; and iron and steel are the top five sectors responsible for GHG emissions in FY2005 and FY2030. This is due to electricity and coal mining, petroleum, and natural gas being related to energy and utility; and fuel combustion and transport and postal services include the fossil fuel combustion stage. The CFP of the iron and steel sector covers the mining operation to the gate stage. Between FY2005 and FY2030 (BAU), input from fuel combustion and coal

mining, petroleum, and natural gas will be reduced, since energy performance will be improved in FY2030. The shift of energy mix will also influence the CFP of the electricity sector, especially due to the share of renewable energy in total electricity production. Moreover, the CFP of electricity in FY2030 (NDC) is estimated to be lower than in FY2030 (BAU) thanks to emission-mitigation technologies including carbon capture and storage (CCS), which will help to achieve the goals of the NDC scenario. From the production-side point of view, in the three scenarios of FY2005, FY2030 (BAU), and FY2030 (NDC), the CFP is reduced by 25, 24, and 19%, respectively, mainly explained by changes in energy sources for electricity generation.

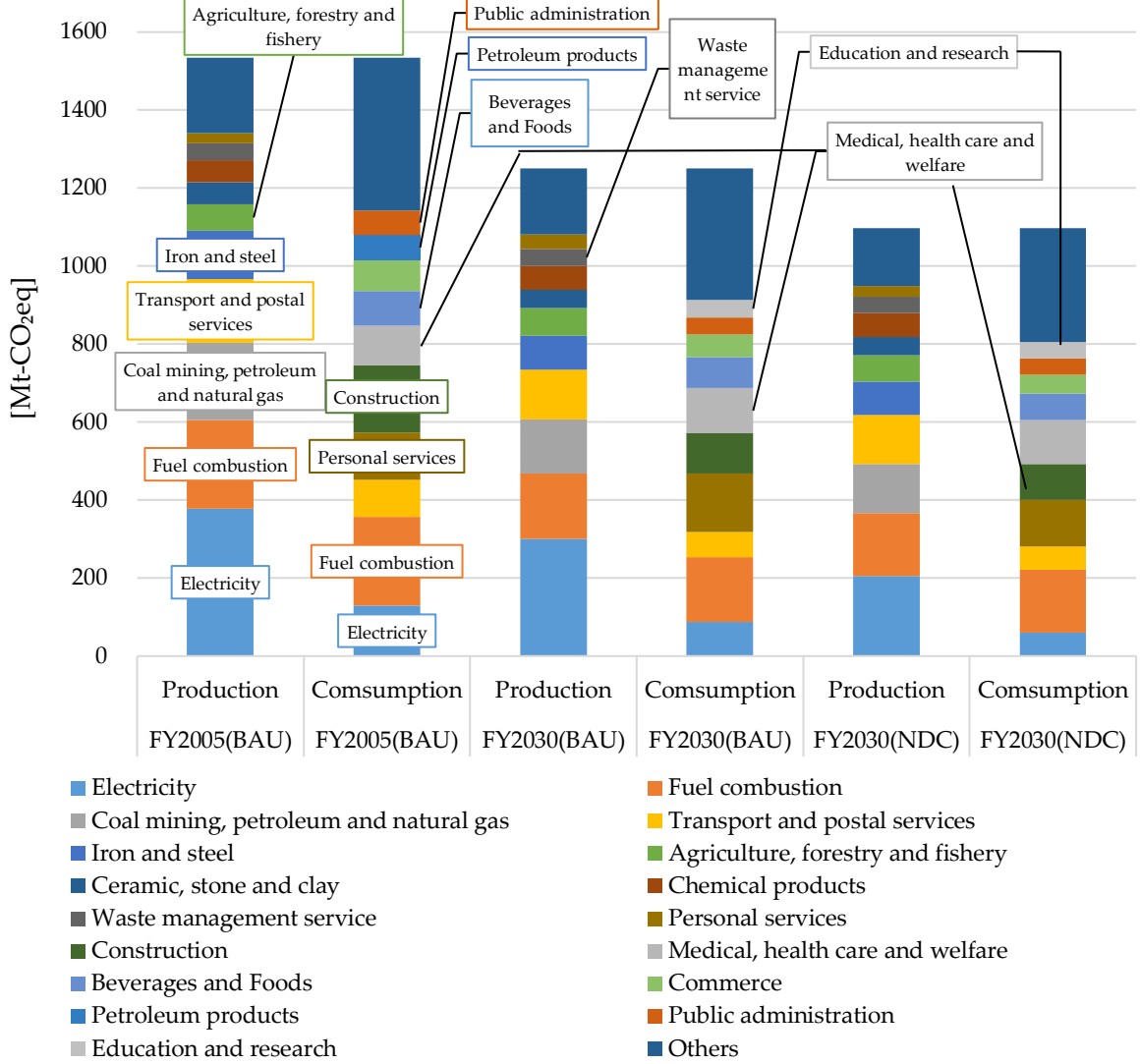

**Figure 4.** Industry-level breakdown from the viewpoint of production and consumption sides.

From the consumption side, fuel combustion, electricity, transport and postal services, personal services, and construction are the top five emission sectors. This is attributed to classification of the CFP from the viewpoint of the consumption side, which allocates the CFP for industrial sectors by taking into account the responsibility of each sector from the purchasing side. In FY2005, petroleum products ranked among the top 10 sectors.

Moreover, the number of elderly people (over 65 years old) in FY2005 was approximately 25 million (20.1% of the total population). In contrast, the number of elderly people in FY2030 is assumed to be 37 million (31.6% of the total population). Hence, the CFP of medical, health, and welfare will increase

in 2030. In addition, personal services will also increase, since this is a prospective sector to support economic growth by, for instance, information technology (IT).

The CFP of construction is related to the fixed capital sector. This is caused by iron and steel and ceramic, stone, and clay as building materials. In addition, the CFP of beverages and foods on the consumption side is caused by agriculture, forestry, and fishery on the production side. Therefore, considering material supply is significant for reduction of the total CFP.

Waste management service is emerging in the top 10 emission sectors as local governments deal with waste management on the production side. In contrast, public administration; medical, health, and welfare; and education and research are among the top 10 emission sectors on the consumption side since they represent the main activities of governments. The total CFP in FY2030 (NDC) is estimated to increase slightly compared with FY2030 (BAU) following the introduction of a carbon budget by the government as one of the global warming countermeasures in this calculation scenario. Thereby, the CFP of the government sector will increase due to its increased final demand in the FY2030 (NDC) scenario.

According to the results, the CFP of the energy and material sectors for iron and steel and transport will have an impact on the total CFP from the production side. From the consumption side, fuel combustion, transport and postal services, and electricity influence the total CFP. Finally, it is important for the reduction of total CFP in the future to cover not only direct impacts from energy use, for example, but also indirect impacts, such as from the material purchase stage.

### 3.2. Carbon Productivity

Figure 5 shows the relationship between the final demand and the CFP of each scenario. The vertical line shows the final demand and the horizontal line shows the CFP of each industrial sector. The CFP for the final demand in each industry is shown using a scatter plot, which indicates carbon productivity. In order to confirm the trend of each scenario, regression lines were drawn. In addition, the broken line shows the regression line to express a relationship between the intensity of final demand and CFP. This intensity is regarded as carbon productivity for each scenario in this figure. The slope of the regression line in each scenario shows the average final demand increase: when one unit of CFP increases, the larger this value, the lower the environmental load of the final demand. In the top left corner of Figure 5, sectors that have a strong influence on economic growth and low emissions are shown, while in the bottom right corner, sectors with a small influence on economic growth and high emissions are shown. Regarding $R^2$, the value for each scenario is within the range $0.4 \lesssim R^2 \lesssim 0.7$, so a certain correlation can be seen between CFP and final demand.

As a result, the CFP shows a tendency to be reduced and final demand to be increased between FY2005 and FY2030. According to these values, carbon productivity will gradually increase and FY2030 (NDC) carbon productivity will be higher than the two other projections. In addition, introduction of the carbon budget in FY2030 (NDC) will cause a slight reduction of final demand in primary and secondary industries. In contrast, the final demand of service sectors in FY2030 (NDC) will be higher than in FY2030 (BAU).

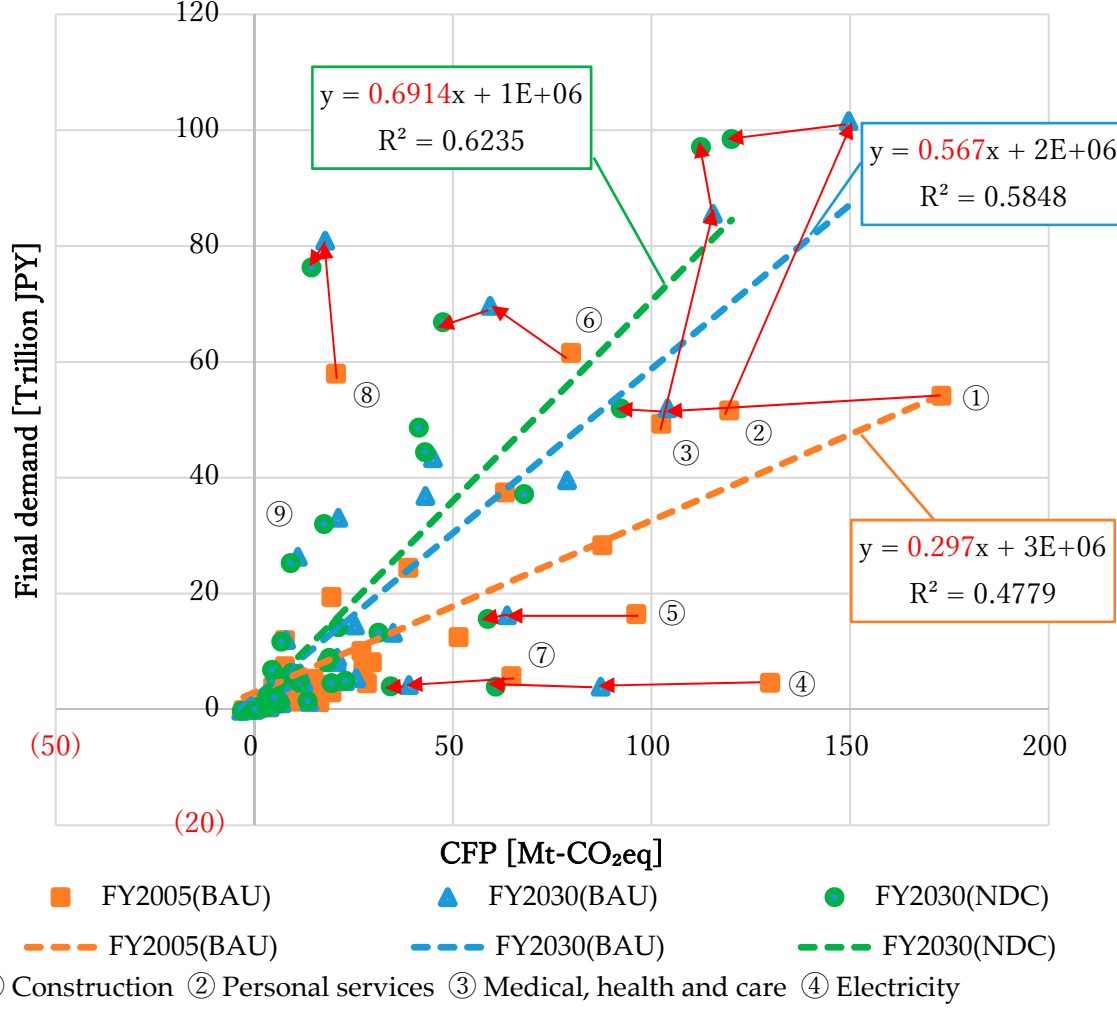

**Figure 5.** Carbon productivity between final demand and CFP.

At the industrial level, the CFP in energy supply sectors, including electricity and petroleum products, will decrease and final demand will remain stable between FY2005 and FY2030. This is attributed to shifts in energy, such as a reduction of fossil fuel and the introduction of renewable energy. Construction is projected to become the highest emission sector, as its final demand is among the top three sectors and direct GHG intensity is also important. Thus, the CFP of construction has a relatively high value as a whole. Commerce has the highest final demand among all sectors, as this sector is heavily linked to all the others. Moreover, the final demand of commerce in FY2030 is accompanied by increased economic growth. The CFP of commerce will decrease in FY2030 due to changes in the energy mix, as this sector needs much electricity to preserve fresh products such as vegetables, fish, and meat as well as for lighting equipment. Personal services will grow between FY2005 and FY2030 with the increased household income supported by GDP growth. Furthermore, this sector has a tendency to purchase much more electricity (e.g., hotels and eating and drinking services, including restaurants). The final demand of medical, health care, and welfare will increase in 2030 due to the aging population. In addition, the final demand of this sector in FY2030 (NDC) will also increase more compared to other sectors, attributed to increased government outcomes following the introduction of the carbon budget. The CFP of transport and postal service will decrease significantly, caused by the transition of fuel transport equipment, including vehicles, switching from gasoline to cleaner energy

such as electricity, for example. Regarding real estate in FY2030, its final demand will be higher than in FY2005. This is due to increased household income with economic growth; in particular, the final demand of household represents a major part of the intermediate sector, indispensable for daily life.

## 4. Discussion

### 4.1. Comparison with Previous Studies

This study estimated CFP projections based on the NDC scenario in 2030 by considering Japan's future economic structure using a combination of the IOA approach and the integrated assessment model AIM/CGE (Japan).

In previous studies, IAMs were used to estimate the total amount of future GHG emissions at global and national levels. On the other hand, they have not been utilized for calculation that details the entire supply chain, where the link between consumption behavior and industrial sectors is shown, such as in the different environmental footprints in the LCA research area. In this study, it was possible to conduct a more detailed analysis by clarifying which consumption behaviors are influenced by which industries. The future values estimated in this study consider both direct and indirect GHG emissions, and the validity can be confirmed when compared with GHG emissions reported in other existing studies.

Oshiro et al. [14] estimated the CFP in Japan for 2030 based on BAU and mitigation (NDC) scenarios as well. For both scenarios, values are about 10% larger in our study due to the inclusion of indirect loads when considering the CFP. However, when looking at the reduction rate between both scenarios in each study, the difference is only about 3% (Table A2), so we assume that the validity of our estimations can be confirmed. Oshiro et al. [27] also estimated GHG emissions, with 2050 as the target year, and it was confirmed that the reduction rate in that calculation was estimated to be about 17% higher compared with our study (Table A2). Indeed, the latter considers as an input parameter 80% GHG emissions reduction by 2050, which as a result has an influence on the results for 2030. Therefore, in order to build a strong comparison with our study, it would more comprehensive to consider 2050 as the target year for both studies.

In order to confirm the reproducibility of the model, we also compared the 2005 database used as a benchmark in our model with another existing database mentioned previously, 3EID (Figure A4). The clear difference with 3EID is the method used to estimate direct emissions per monetary unit (Japanese yen) for each industry. In 3EID, fuel consumption (e.g., oil) is an input for each industry, estimated through statistical data. On the other hand, in our study, the amount of fuel input (in kg) for each industry is calculated directly from the purchase price (in Japanese yen). Indeed, it is difficult to estimate these inputs using a physical quantity, as an economic model is used to estimate future inputs. However, it must be noted that in our model adjustments are also made by imposing a constraint on the calculation so that the total amount does not exceed the GHG emissions reported by NIR. Therefore, the total amount is the same, however the direct GHG emissions for each industrial sector are different. As a result of comparison, it was confirmed that the GHG emissions for each industry were within a margin of error of about 5%.

Finally, the 2030 (BAU) scenario based on the 2005 standard cannot achieve the target reduction of 25.4%, but this can be achieved in the 2030 (NDC) scenario when global warming countermeasures are taken.

### 4.2. Limitations

This study estimated the JIOT in 2030 using AIM/CGE (Japan). Import products in this table are assumed to have the same proprieties as the technology in Japan. Thus, the CFP of products, which are sometimes imported from countries where the technology performance is lower or higher, is different than the real expected value, as this study considers the same direct GHG emissions for all of these products. To address this limitation, it would be better to use a multi-regional input–output (MRIO)

table such as EXIOBASE as well as an IAM that covers the global scale. However, another limitation in that case would be that the number of sectors in the MRIO table is usually lower than in a national-level IOT such as the JIOT.

In the NDC scenario, the cost of energy shift, including the introduction of renewable energy instead of thermal power generation, was included to estimate the CFP projection. On the other hand, material consumption could not be considered specifically, such as how much silicon will be needed to provide solar power generation. In addition, we also need to think about how much capacity is needed to introduce renewable energy plants. Finally, in the future it would be better to estimate not only the CFP, but also the environmental impact of other factors such as material consumption and land use.

## 5. Conclusions

We estimated the CFP projection using AIM/CGE (Japan) as an integrated assessment model. In this study, the CFP projection for FY2030 (NDC) was estimated to be reduced by 28% compared to 2005. Moreover, the CFP will be reduced by 25.4% in the NDC-based scenario results from the Japanese government. Therefore, we consider our results to be valid. It is possible to interpret the results considering each final demand by a breakdown of the industrial sector level. CFP per capita was estimated to be reduced by 23% in this study. As a result, energy shift will contribute to reducing CFP in the future. Therefore, it is important to consider reducing the input amounts of primary products such as coal and petroleum and encourage a shift of the electricity mix from nonrenewable to renewable energy.

These types of countermeasures have also been proposed in other studies. Moreover, our results indicate that the construction sector is also important when considering reducing the CFP without considering an energy shift. In particular, considering the building materials circularity such as for iron and steel would contribute to CFP reduction in 2030.

By considering both production-side and consumption-side approaches, it is shown that each side has responsibility for GHG emissions, so it is necessary for governments not only to establish laws or regulations but also to inform and discuss with citizens the possible impacts of climate change in order to change behaviors. Such inclusion has been proposed recently at the city level in the Climate Emergency Declaration (CED).

The CFP projection is also influenced by the age distribution of the population (e.g., elderly people), and the introduction of a carbon budget will also have a positive effect on reducing the total CFP. As the number of elderly people is expected to continue increasing in Japan in the future (about one-third of the population is predicted to be above 60 years old in 2050), it is necessary for policy-makers to tackle the subject as soon as possible.

In the future, it would be better to consider the environmental impact of other factors, such as material consumption and land use.

**Author Contributions:** Conceptualization, Y.I., T.M., and N.I.; Methodology, Y.I., T.M., and N.I.; Formal Analysis, Y.I. and T.M.; Investigation, Y.I. and T.M.; Writing—Original Draft Preparation, Y.I.; Supervision, N.I.; Writing—Review & Editing, Y.I., T.M., S.K., and N.I.

**Funding:** This research received no external funding.

**Conflicts of Interest:** The authors declare no conflict of interest.

# Appendix A

**Table A1.** Summary of previous integrated assessment models.

| Model Name | Institution | Area, Output | Application |
|---|---|---|---|
| Asia-Pacific Integrated Model (AIM) | National Institute for Environmental Science (NIES), Japan | Asia-Pacific, national economic level, GHG emissions | Akimoto et al. (2015) [14] |
| Dynamic Integrated model of Climate and Economy (DICE) | Yale University, USA | Global, industrial and land use $CO_2$, anthropogenic emissions | Su et al. (2017) [28] |
| Global Charge Assessment Model (GCAM) | Joint Global Change Research Institute, USA | 32 geopolitical regions, global primary energy, price of $CO_2$ per ton | Thomson et al. (2011) [29] |
| Integrated Model to Assess the Global Environment (IMAGE) | PBL Netherlands Environmental Assessment Agency, Netherlands | Global and national economic levels, $CO_2$ equivalent, land use | Strengers et al. (2008) [30] |
| Model for Energy Supply Strategy Alternatives and Their General Environmental Impact (MESSAGE) | International Institute for Applied Systems Analysis, Austria | Global and national economic levels, global $CO_2$ emissions, land-use and land-cover change | Keywan et al. (2011) [31] |

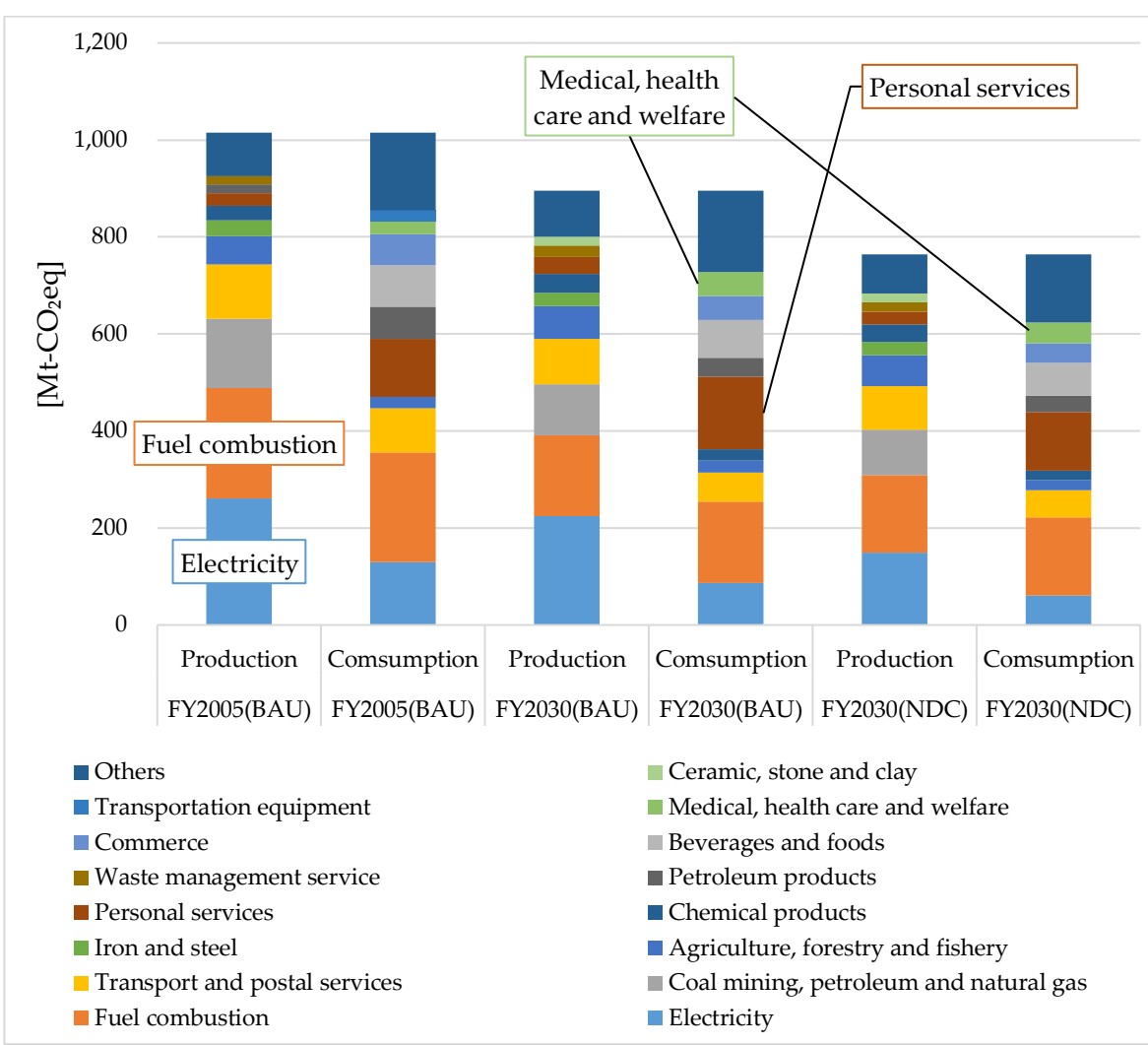

**Figure A1.** Industrial level breakdown from the viewpoint of production and consumption sides for households.

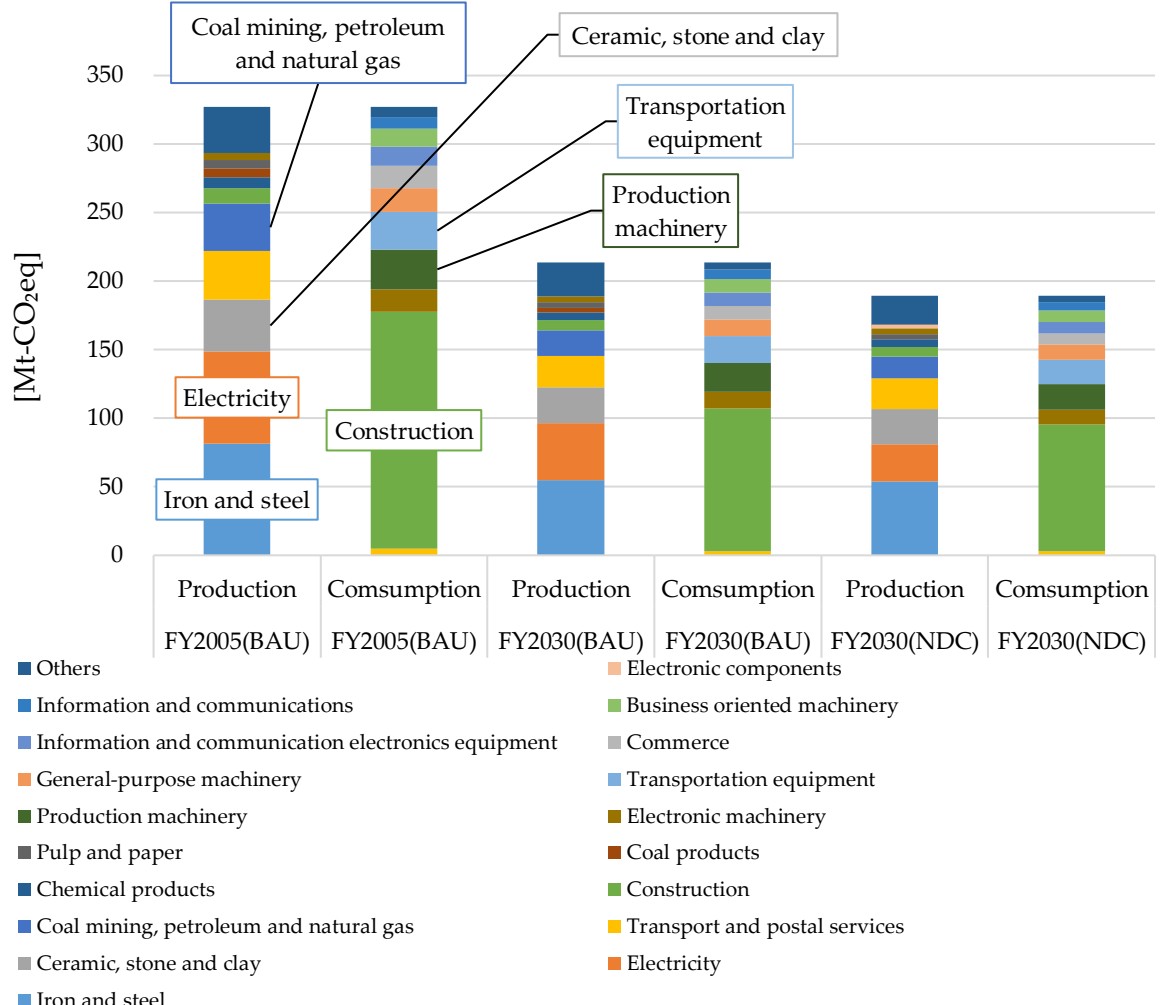

**Figure A2.** Industrial level breakdown from the viewpoint of production and consumption side in fixed capital.

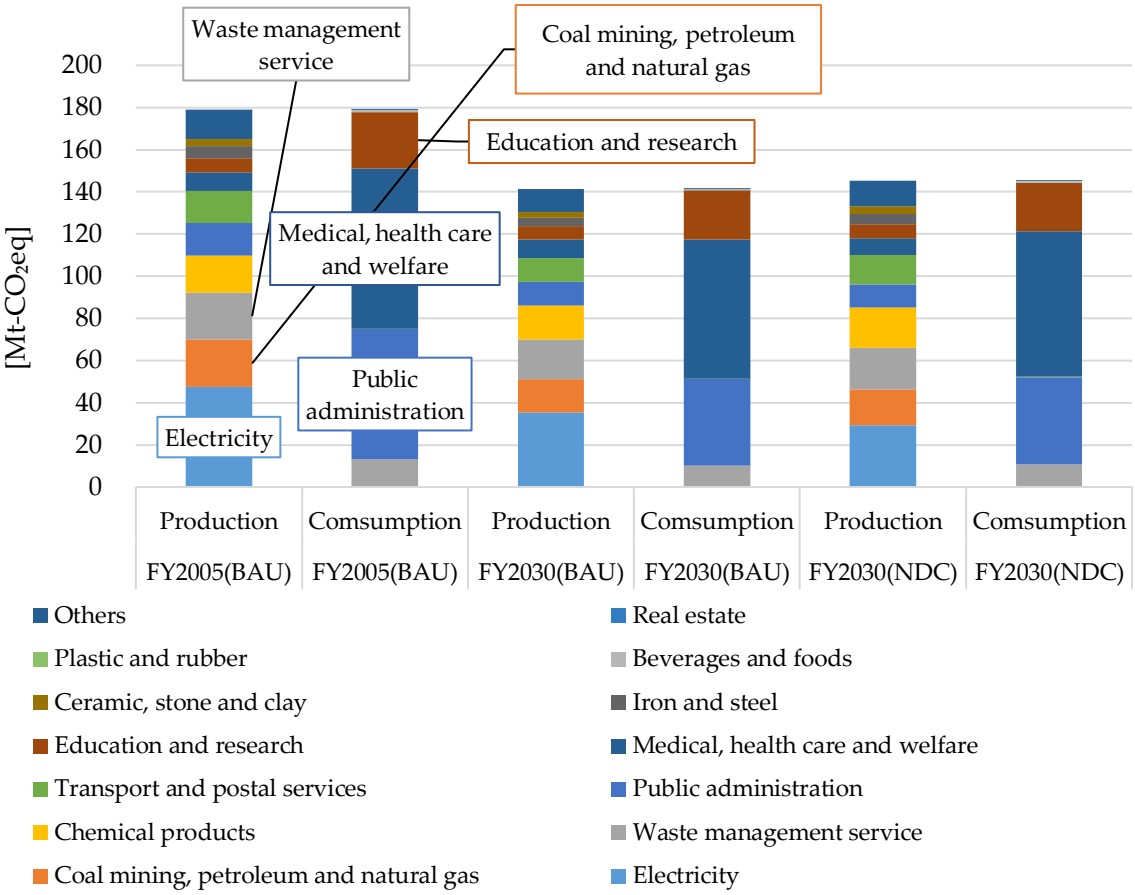

**Figure A3.** Industrial level breakdown from the viewpoint of production and consumption sides in government.

**Table A2.** Comparison of CFP in 2030 between this study and previous studies.

| Index | This Study | Oshiro et al. [13] | Oshiro et al. [27] |
|---|---|---|---|
| BAU (MtCO$_2$eq) | 1251 | 1130 | 1160 |
| Mitigation (tCO$_2$eq) | 1098 | 960 | 821 |
| GHG ratio (%) | −12 | −15 | −29 |

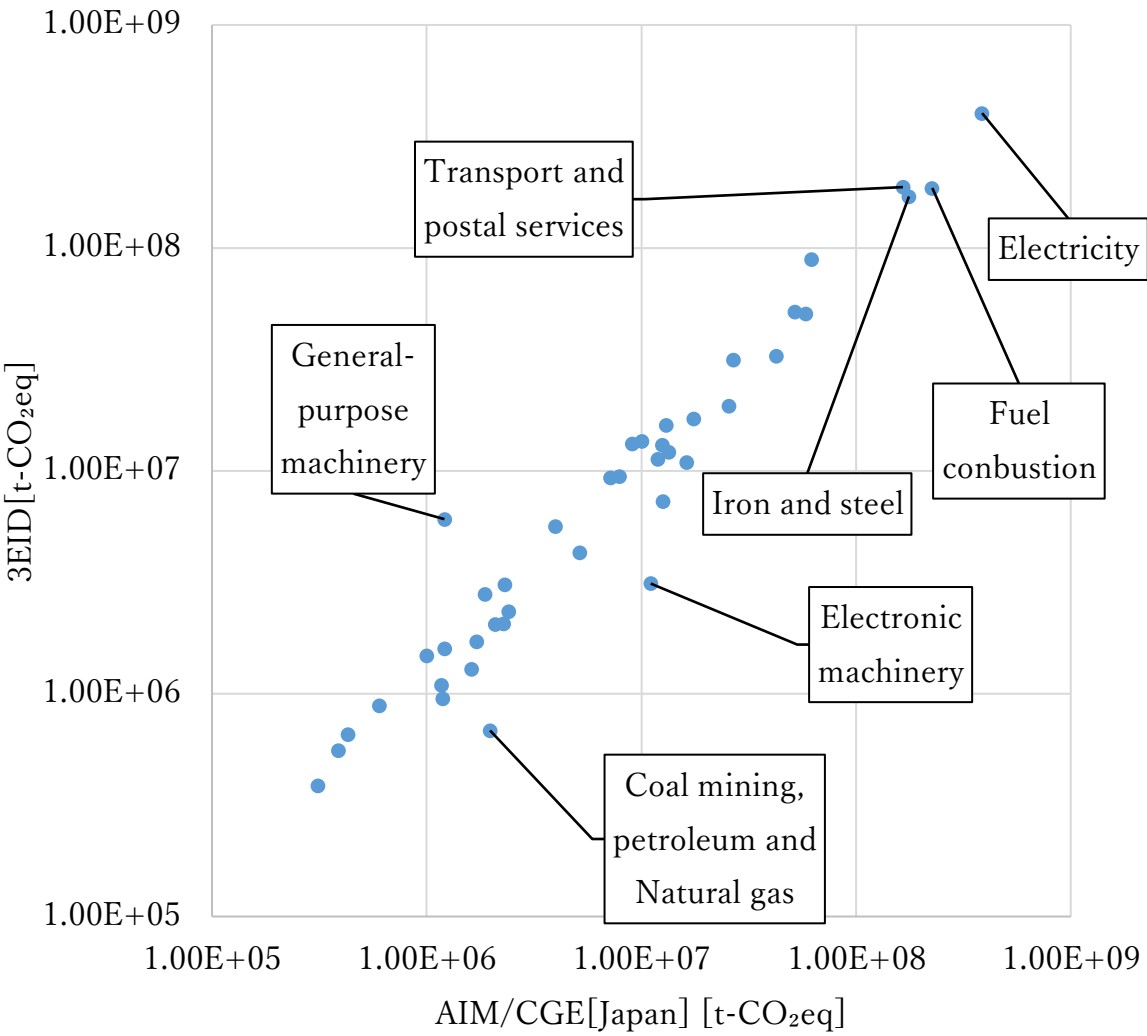

**Figure A4.** Comparison of direct GHG emissions between this study and Embodied Energy and Emission Intensity Data for Japan Using Input–Output Tables (3EID) [23].

**Table A3.** CFP results of industrial level breakdown (MtCO₂eq) (Color display as follow: red: high, green: low).

| Scenario | Sectors | Total | Household | Fixed Capital | Government |
|---|---|---|---|---|---|
| FY2005(BAU) | Agriculture, forestry and fishery | 23.8 | 22.6 | 1.3 | 0.0 |
| FY2005(BAU) | Mining | −0.1 | −0.1 | −0.0 | 0.0 |
| FY2005(BAU) | Coal mining, petroleum and natural gas | 0.0 | 0.0 | 0.0 | 0.0 |
| FY2005(BAU) | Beverages and foods | 86.9 | 85.9 | 0.0 | 1.0 |
| FY2005(BAU) | Textile products | 13.2 | 12.3 | 0.8 | 0.0 |
| FY2005(BAU) | Pulp and paper | 4.3 | 2.7 | 1.6 | 0.0 |
| FY2005(BAU) | Chemical products | 18.9 | 18.9 | 0.0 | 0.0 |
| FY2005(BAU) | Petroleum products | 66.8 | 66.8 | 0.0 | 0.0 |
| FY2005(BAU) | Coal products | 0.0 | 0.0 | 0.0 | 0.0 |
| FY2005(BAU) | Plastic and rubber | 6.3 | 6.3 | −0.0 | 0.0 |
| FY2005(BAU) | Ceramic, stone and clay | 3.2 | 3.2 | 0.0 | 0.0 |
| FY2005(BAU) | Iron and steel | −4.1 | −0.6 | −3.5 | 0.0 |
| FY2005(BAU) | Non-ferrous metals | 0.5 | 0.4 | 0.1 | 0.0 |
| FY2005(BAU) | Metal products | 3.7 | 2.1 | 1.6 | 0.0 |
| FY2005(BAU) | General-purpose machinery | 17.2 | 0.1 | 17.2 | 0.0 |
| FY2005(BAU) | Production machinery | 29.0 | 0.1 | 28.9 | 0.0 |
| FY2005(BAU) | Business oriented machinery | 14.9 | 1.7 | 13.2 | 0.0 |

**Table A3.** *Cont*.

| Scenario | Sectors | Total | Household | Fixed Capital | Government |
|---|---|---|---|---|---|
| FY2005(BAU) | Electronic components | 0.9 | 0.9 | 0.0 | 0.0 |
| FY2005(BAU) | Electronic machinery | 27.3 | 10.9 | 16.4 | 0.0 |
| FY2005(BAU) | Information and communication electronics equipment | 27.2 | 13.3 | 13.9 | 0.0 |
| FY2005(BAU) | Transportation equipment | 50.4 | 23.0 | 27.4 | 0.0 |
| FY2005(BAU) | Miscellaneous manufacturing products | 7.4 | 4.8 | 2.6 | 0.0 |
| FY2005(BAU) | Construction | 173.1 | 0.0 | 173.1 | 0.0 |
| FY2005(BAU) | Electricity | 129.9 | 129.9 | 0.0 | 0.0 |
| FY2005(BAU) | Gas supply | 11.4 | 11.4 | 0.0 | 0.0 |
| FY2005(BAU) | Steam and hot water supply | 0.1 | 0.1 | 0.0 | 0.0 |
| FY2005(BAU) | Water supply | 6.7 | 6.7 | 0.0 | 0.0 |
| FY2005(BAU) | Waste management service | 16.4 | 3.2 | 0.0 | 13.2 |
| FY2005(BAU) | Commerce | 79.5 | 63.0 | 16.6 | 0.0 |
| FY2005(BAU) | Finance and insurance | 7.8 | 7.8 | 0.0 | 0.0 |
| FY2005(BAU) | Real estate | 20.6 | 20.6 | 0.0 | 0.0 |
| FY2005(BAU) | Transport and postal services | 95.8 | 91.0 | 4.7 | 0.0 |
| FY2005(BAU) | Information and communications | 19.5 | 11.0 | 8.4 | 0.0 |
| FY2005(BAU) | Public administration | 63.1 | 1.3 | 0.0 | 61.8 |
| FY2005(BAU) | Education and research | 38.8 | 12.1 | 0.0 | 26.7 |
| FY2005(BAU) | Medical, health care and welfare | 102.5 | 26.4 | 0.0 | 76.1 |
| FY2005(BAU) | Miscellaneous non-profit services | 4.8 | 4.8 | 0.0 | 0.0 |
| FY2005(BAU) | Business services | 7.7 | 4.8 | 2.9 | 0.0 |
| FY2005(BAU) | Personal services | 119.6 | 119.6 | 0.0 | 0.0 |
| FY2005(BAU) | Office supplies and activities not elsewhere classified | 0.1 | 0.1 | 0.0 | 0.0 |
| FY2005(BAU) | Fuel combustion | 226.5 | 226.5 | | |
| FY2030(BAU) | Agriculture, forestry and fishery | 25.7 | 24.6 | 1.1 | 0.0 |
| FY2030(BAU) | Mining | −0.1 | −0.1 | −0.0 | 0.0 |
| FY2030(BAU) | Coal mining, petroleum and natural gas | 0.0 | 0.0 | 0.0 | 0.0 |
| FY2030(BAU) | Beverages and foods | 78.8 | 78.1 | 0.0 | 0.6 |
| FY2030(BAU) | Textile products | 11.7 | 11.1 | 0.6 | 0.0 |
| FY2030(BAU) | Pulp and paper | 4.2 | 3.0 | 1.2 | 0.0 |
| FY2030(BAU) | Chemical products | 22.8 | 22.8 | 0.0 | 0.0 |
| FY2030(BAU) | Petroleum products | 38.9 | 38.9 | 0.0 | 0.0 |
| FY2030(BAU) | Coal products | 0.0 | 0.0 | 0.0 | 0.0 |
| FY2030(BAU) | Plastic and rubber | 6.1 | 6.1 | −0.0 | 0.0 |
| FY2030(BAU) | Ceramic, stone and clay | 3.8 | 3.8 | 0.0 | 0.0 |
| FY2030(BAU) | Iron and steel | −3.3 | −0.6 | −2.7 | 0.0 |
| FY2030(BAU) | Non-ferrous metals | 0.6 | 0.6 | 0.1 | 0.0 |
| FY2030(BAU) | Metal products | 2.6 | 1.5 | 1.1 | 0.0 |
| FY2030(BAU) | General-purpose machinery | 12.4 | 0.1 | 12.3 | 0.0 |
| FY2030(BAU) | Production machinery | 20.9 | 0.1 | 20.7 | 0.0 |
| FY2030(BAU) | Business oriented machinery | 11.6 | 2.0 | 9.6 | 0.0 |
| FY2030(BAU) | Electronic components | 1.0 | 1.0 | 0.0 | 0.0 |
| FY2030(BAU) | Electronic machinery | 20.7 | 8.3 | 12.4 | 0.0 |
| FY2030(BAU) | Information and communication electronics equipment | 25.3 | 15.2 | 10.1 | 0.0 |
| FY2030(BAU) | Transportation equipment | 35.0 | 15.3 | 19.6 | 0.0 |
| FY2030(BAU) | Miscellaneous manufacturing products | 6.8 | 5.0 | 1.8 | 0.0 |
| FY2030(BAU) | Construction | 104.0 | 0.0 | 104.0 | 0.0 |
| FY2030(BAU) | Electricity | 87.2 | 87.2 | 0.0 | 0.0 |
| FY2030(BAU) | Gas supply | 6.9 | 6.9 | 0.0 | 0.0 |
| FY2030(BAU) | Steam and hot water supply | 0.0 | 0.0 | 0.0 | 0.0 |
| FY2030(BAU) | Water supply | 6.7 | 6.7 | 0.0 | 0.0 |
| FY2030(BAU) | Waste management service | 13.9 | 3.7 | 0.0 | 10.2 |
| FY2030(BAU) | Commerce | 59.4 | 49.9 | 9.5 | 0.0 |
| FY2030(BAU) | Finance and insurance | 11.0 | 11.0 | 0.0 | 0.0 |
| FY2030(BAU) | Real estate | 17.9 | 17.9 | 0.0 | 0.0 |
| FY2030(BAU) | Transport and postal services | 63.6 | 60.5 | 3.1 | 0.0 |
| FY2030(BAU) | Information and communications | 21.2 | 14.2 | 7.0 | 0.0 |
| FY2030(BAU) | Public administration | 43.1 | 2.0 | 0.0 | 41.1 |
| FY2030(BAU) | Education and research | 45.0 | 21.7 | 0.0 | 23.3 |

**Table A3.** *Cont.*

| Scenario | Sectors | Total | Household | Fixed Capital | Government |
|---|---|---|---|---|---|
| FY2030(BAU) | Medical, health care and welfare | 115.5 | 49.5 | 0.0 | 66.0 |
| FY2030(BAU) | Miscellaneous non-profit services | 5.2 | 5.2 | 0.0 | 0.0 |
| FY2030(BAU) | Business services | 8.0 | 5.9 | 2.1 | 0.0 |
| FY2030(BAU) | Personal services | 149.7 | 149.7 | 0.0 | 0.0 |
| FY2030(BAU) | Office supplies and activities not elsewhere classified | 0.1 | 0.1 | 0.0 | 0.0 |
| FY2030(BAU) | Fuel combustion | 167.0 | 167.0 | - | - |
| FY2030(NDC) | Agriculture, forestry and fishery | 23.0 | 22.0 | 1.0 | 0.0 |
| FY2030(NDC) | Mining | −0.1 | −0.0 | −0.0 | 0.0 |
| FY2030(NDC) | Coal mining, petroleum and natural gas | 0.0 | 0.0 | 0.0 | 0.0 |
| FY2030(NDC) | Beverages and foods | 68.0 | 67.3 | 0.0 | 0.7 |
| FY2030(NDC) | Textile products | 9.7 | 9.2 | 0.5 | 0.0 |
| FY2030(NDC) | Pulp and paper | 3.4 | 2.4 | 1.0 | 0.0 |
| FY2030(NDC) | Chemical products | 19.5 | 19.5 | 0.0 | 0.0 |
| FY2030(NDC) | Petroleum products | 34.4 | 34.4 | 0.0 | 0.0 |
| FY2030(NDC) | Coal products | 0.0 | 0.0 | 0.0 | 0.0 |
| FY2030(NDC) | Plastic and rubber | 4.9 | 4.9 | −0.0 | 0.0 |
| FY2030(NDC) | Ceramic, stone and clay | 3.1 | 3.1 | 0.0 | 0.0 |
| FY2030(NDC) | Iron and steel | −3.1 | −0.6 | −2.6 | 0.0 |
| FY2030(NDC) | Non-ferrous metals | 0.5 | 0.5 | 0.1 | 0.0 |
| FY2030(NDC) | Metal products | 2.4 | 1.4 | 1.0 | 0.0 |
| FY2030(NDC) | General-purpose machinery | 11.3 | 0.1 | 11.2 | 0.0 |
| FY2030(NDC) | Production machinery | 18.9 | 0.1 | 18.8 | 0.0 |
| FY2030(NDC) | Business oriented machinery | 10.0 | 1.6 | 8.4 | 0.0 |
| FY2030(NDC) | Electronic components | 0.9 | 0.9 | 0.0 | 0.0 |
| FY2030(NDC) | Electronic machinery | 18.3 | 7.3 | 11.0 | 0.0 |
| FY2030(NDC) | Information and communication electronics equipment | 21.1 | 12.5 | 8.6 | 0.0 |
| FY2030(NDC) | Transportation equipment | 31.3 | 13.8 | 17.5 | 0.0 |
| FY2030(NDC) | Miscellaneous manufacturing products | 5.7 | 4.1 | 1.6 | 0.0 |
| FY2030(NDC) | Construction | 92.3 | 0.0 | 92.3 | 0.0 |
| FY2030(NDC) | Electricity | 60.8 | 60.8 | 0.0 | 0.0 |
| FY2030(NDC) | Gas supply | 6.4 | 6.4 | 0.0 | 0.0 |
| FY2030(NDC) | Steam and hot water supply | 0.0 | 0.0 | 0.0 | 0.0 |
| FY2030(NDC) | Water supply | 5.0 | 5.0 | 0.0 | 0.0 |
| FY2030(NDC) | Waste management service | 13.4 | 2.7 | 0.0 | 10.8 |
| FY2030(NDC) | Commerce | 47.5 | 39.6 | 7.9 | 0.0 |
| FY2030(NDC) | Finance and insurance | 9.2 | 9.2 | 0.0 | 0.0 |
| FY2030(NDC) | Real estate | 14.4 | 14.4 | 0.0 | 0.0 |
| FY2030(NDC) | Transport and postal services | 58.8 | 55.8 | 3.0 | 0.0 |
| FY2030(NDC) | Information and communications | 17.6 | 11.6 | 6.0 | 0.0 |
| FY2030(NDC) | Public administration | 43.0 | 1.6 | 0.0 | 41.4 |
| FY2030(NDC) | Education and research | 41.4 | 18.2 | 0.0 | 23.3 |
| FY2030(NDC) | Medical, health care and welfare | 112.5 | 43.5 | 0.0 | 68.9 |
| FY2030(NDC) | Miscellaneous non-profit services | 4.5 | 4.5 | 0.0 | 0.0 |
| FY2030(NDC) | Business services | 6.8 | 5.0 | 1.8 | 0.0 |
| FY2030(NDC) | Personal services | 120.2 | 120.2 | 0.0 | 0.0 |
| FY2030(NDC) | Office supplies and activities not elsewhere classified | 0.1 | 0.1 | 0.0 | 0.0 |
| FY2030(NDC) | Fuel combustion | 160.7 | 160.7 | - | - |

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
