# Peer review of "Projection of National Carbon Footprint in Japan with Integration of LCA and IAMs"

_sustainability, doi:10.3390/su11236875_

Round 1

Reviewer 1 Report

Dear authors,

Congratulations for the theoretical and empirical high quality of the article. I was particular driven by the detail of the analysis carry on and the results achieved, which I found of the most scientific soundness. Having aid that, It's my believe that conclusions are less rich if compared with the meticulous and hard operational work executed. I would advice to add a few line addressing study ambition (the advance of your study could provide beyond the state-of-art and the extent of your ambitious when you mention results "...support decision making..."). Moreover, something I do regret is the lacking of landuse land cover data which is critical.

Last comment, to call your attention for some conceptual fuzziness upon BAU Scenario and Baseline scenario. I haven't quite get if 2005 was a baseline scenario or a BAU scenario, as 2030! I would like some clarification regarding scenarios

Reviewer 2 Report

The paper presents estimations regarding the Japan's national carbon footprint, under a baseline and two future scenarios (for 2030). 

It is a very interesting idea and for the estimations modelling and life cycle assessment are applied. The outcome, is interesting for the policy makers and relevant stakeholders of Japan, as well as all over the world.

However, there are several points that need to be addressed before accepting it for publication. These are:

1) It should clearly present novelty and originality, since the results seem similar to the reference 25. If these calculations for Japan are already there, then which exactly is the novelty of this work and why it should be published?

2) In the introduction, a deeper literature review should be done a) to inform readers about similar approaches and results for other areas of the world and b) to highlight the novelty of this work. In my view, the text know focuses on modelling approaches and info about the importance of this work is missing. Also, the table is not appropriate there, should be placed as supplementary material.

3) The materials and methods should include some more info about the FY(2030) scenario. In Figure 1, why the inputs for FY2005 and FY2030 are different? In line 162, GHG emissions do not include "high-fructose corn syrup" (Hydro-Fluoro-Carbons maybe?). See lines 123-4, in support of comment 1: If such data are already available, what is the aim of this research and which is the novelty? Are there any of the emission factors (e.g. IPCC) used for your calculations? Cite the sources for those data. In the results, you have a section about Carbon productivity. Where is the respective text in the materials and methods? (maybe I missed it but the sections should have consistency). 

4) There are too many figures in the results, try to reduce it and do not repeat what is presented there in the text. Correlations (e.g. Figure 7) should be accompanied with statistical significance data. Also, seems that these two figures (7a,b) repeat what already presented.

5) The discussion section usually does not contain tables and figures, these belong to the results. Here, it is obvious that your results are similar to other studies in Japan. If yes, then we go to point 1, on novelty again. 

Overall, in my view the above should be adressed at this point and additionally, the manuscript shoould be significantly improved, having in mind the broad readership of sustainability journal. 

Round 2

Reviewer 2 Report

I would like to thank the editor for this opportunity and also the authors for considering my comments, which targeted to the improvement of their work. 

In my opinion, the manuscript is clearly improved, the methods are sound, the results are sufficiently presented and discussed and the novelty of this work is highlighted. 

It is a valuable paper that could be used as a reference for such assessments in other countries or areas, towards GHG emissions reporting and mitigation.

My only comment however, although I am not a statistics expert, is that in Figure 5, the statistical significance of the correlations should be presented. 

There are minor grammar and syntax errors e.g.

line 10: projections have...

25: due to the introduction

36: GHG emissions have been paid attention more

100: estimate

105: introduced

206: according NOT thanks...

291: slight reduction

331: larger than in our study

335 compared to..

Therefore, it is good to ask a native speaker correct the manuscript before publication. Other than these, my recommendation is to publish this paper. 

Author Response

Dear Reviewer 2

Thank you very much for your precious comments which helped to improve our manuscript.

We have highlighted the statistical significance of the correlation (lines 284-293) and got the grammar check by native speaker as you recommended.